# Satisfied and Frustrated Needs, Subjective Vitality and University Students' Life Satisfaction of Physical Activity and Sports

Heriberto Antonio Pineda-Espejel [1,*] , Raquel Morquecho-Sánchez [2], Lucía Terán [1], Icela López-Gaspar [3], Antonio Hernández-Mendo [4] , Verónica Morales-Sánchez [4], Encarnación Chica-Merino [5] and Antonio Granero-Gallegos [6]

1 Facultad de Deportes, Universidad Autónoma de Baja California, Mexicali 21000, Mexico
2 Facultad de Organización Deportiva, Universidad Autónoma de Nuevo León, Monterrey 64460, Mexico
3 Facultad de Idiomas, Universidad Autónoma de Baja California, Mexicali 21000, Mexico
4 Department of Social Psychology, Social Work, Social Anthropology and East Asian Studies, University of Malaga, 29071 Malaga, Spain
5 SAFA University Center, University of Jaén, Avenida Cristo Rey, 17, 23400 Jaén, Spain
6 Department of Education, Faculty of Education Sciences, Universidad de Almería, 04120 Almería, Spain
* Correspondence: antonio.pineda@uabc.edu.mx

**Abstract:** This study is based on frameworks of the eudaimonic activity model and the basic psychological needs theory, with two purposes: one, to prove the validity of a translation and adaptation of the Basic Psychological Needs Satisfaction and Frustration Scale; two, to analyze how the basic psychological satisfaction and frustration needs influences the well-being of university physical activity and sports students, through sex. A total of 830 University students of physical activity and sports with an age between 17 and 31 years (M = 20.70 years; ±2.96) participated. The sample was divided into two subsamples by random selection of 50% of the cases, preserving the relative distribution of sex and age. The first subsample was used to validate the adaptation of the Basic Psychological Need Satisfaction and Frustration Scale to Spanish as spoken in Mexico (Study 1); and the second subsample was used to test a proposed sequential theoretical model (Study 2). In Study 1, the CFA supported the structure of six factors—satisfaction of needs for autonomy, relatedness and competence; frustration of needs for autonomy, relatedness and competence (RMSR = 0.04; RMSEA = 0.046; TLI = 0.93; CFI = 0.94)—as well as the structure of six first-order factors plus two second-order factors—psychological need satisfaction and psychological need frustration (RMSR = 0.05; RMSEA = 0.055; TLI; CFI = 0.91). Both structures were equivalent between men and women. In Study 2, the results of the structural equations model show good fit (RMSEA = 0.05; TLI = 0.90; CFI = 0.92), indicating that the needs satisfied and frustrated contribute in a unique way to indicators of eudaimonic well-being (i.e., subjective vitality) and subjective well-being (i.e., life satisfaction), being equivalent through sex. In conclusion, satisfaction of competence, relationships and autonomy are essential nutrients for a positive performance in this sample.

**Keywords:** basic psychological needs; psychometric properties; psychological well-being; sport

## 1. Introduction

Ryan and Deci [1] established the organization of well-being studies into two traditions. One is basically associated with happiness (hedonic well-being), and the other is related to human potential development (eudaimonic well-being). Both are relevant in sports and physical activity because they require planning objectives, setting goals and adapting to changes, as well as high volition.

Eudaimonic well-being is related to human potential [1], focusing on meaning and self-realization. Its definition is based on the degree to which a person is fully functional and

achieving personal growth. The Eudaimonic Activity Model (EAM) [2–4] proposes a wider well-being construct, in which we can distinguish "doing something good" from "feeling good". Within "feeling good", we have on one side, functioning eudaimonic aspects such as the basic psychological needs [5]; on the other, aspects of subjective well-being (SWB).

Self-determination theory (SDT) [6,7] is a theoretical framework of human development and well-being [8], and the basic psychological needs theory emanates from it (BPNT) [9]. The aforementioned theory is a relevant framework that takes into consideration the bright and dark sides of people's behavior [10]; it points out that there are "specific psychological and social nutrients that, when satisfied within the interpersonal and cultural contexts of the development of individuals, enable psychological growth, integrity and well-being" [8] (p. 82).

BPNT specifies three innate psychological needs: competence (i.e., feelings of efficiency at the moment of interacting with the environment, and a feeling of control, efficacy and trust in actions taken), autonomy (i.e., sensation of will and of being the master of one´s own conduct) and relatedness (i.e., feelings of love, of being related to others and of caring for significant others). These basic psychological needs (BPN), as psychological experiences, are universal across cultures and people and they apply to all aspects of life [11].

Whereas the satisfaction of the BPN is required for proper health and functioning across individuals, resulting in optimal human functioning and personal well-being, the frustration of BPN contributes to uneasiness and to decreasing personal functioning [9]. Satisfaction is considered the way by which development and optimal functioning versus passivity can be understood. Conversely, frustration is suffered when people´s psychological needs are not only unsatisfied, but actively undermined by others, making the person feel incompetent, isolated and controlled by others [10,12].

Several researchers have questioned whether a low score in satisfaction of the BPN can reflect the active nature and the intensity of needs frustration [12]. Bartholomew et al. [12] indicated that the satisfaction of the BPN is commonly measured using items that only collect positive psychological experiences (e.g., feelings of support, understanding); thus, they probably do not collect the negative aspects of these experiences (e.g., contempt, conflict, feelings of rejection).

On the other hand, subjective vitality has been considered an indicator of eudaimonic well-being [1], and it refers to the conscious experience of feeling alive and energetic, which is perceived to originate in the self [13]. It means that the person is in line with his/her self-values and needs [14]. People differ in their experience of subjective vitality in function not only of physical influences (e.g., states of illness and fatigue), but of some psychological factor, such as that of being effective [13], linked to the need of competence. In this respect, subjective vitality can be decreased by factors that block or hinder the needs of competence or autonomy [13]. Thus, the distinction between satisfaction and frustration of BPN is crucial, since both separate experiences are related with different consequences [8].

As for SWB, it is defined as "a wide category of phenomena which include the emotional responses of people, the satisfaction with the environments and the global judgements of life-satisfaction" [15] (p. 277) referring to general and out of context feelings and to positive or negative life evaluations [2]. In this respect, SWB covers cognitive and emotional evaluations that an individual makes regarding to his/her life. The experience of life-satisfaction is included within the cognitive evaluations [15,16].

For Veenhoven et al. [17], life-satisfaction is the degree in which a person positively evaluates the general quality of his/her life as a whole, or within specific life environments (e.g., family life, school experiences) [18,19]. A high life-satisfaction suggests that the quality of life is good; a low life-satisfaction indicates a serious shortage of some kind [20].

## 2. Current Study

VanderWeele et al. [21] emphasized that for a better understanding of people's well-being, it is necessary to measure multiple aspects of psychological well-being. The general life evaluations of people can serve as primary indicators of how well a person feels, and

the degree of BPN gives us information of why people feel well [2]. For this reason, it is important to measure the BPN to better understand people´s well-being [22].

BPN are especially important when the goal is to measure human flourishing, defined as complete well-being, where all aspects of a person´s life are good [23]. Martela and Ryan [22] mentioned that, currently, the most applied and validated instrument to measure the satisfaction/frustration of BPN is the Basic Psychological Need Satisfaction and Frustration Scale (BPNSFS) [24]. It has been used in populations of university students, workers, unemployed people, teenagers, gym members, adolescent and young students, in contexts such as the Portuguese [25], Vietnamese [26], Arabic [27], Indonesian [28] and French [29], and it demonstrated good reliability and validity.

There are some Spanish versions that have demonstrated adequate psychometric properties, such as those of Chen et al. [24] with a Peruvian subsample, Del Valle et al. [30] with a Chilean sample and by Cardella et al. [31] in a Spanish sample. Evidence of factorial invariance by sex was given only by the last mentioned. There is also a Spanish version adapted in physical education and applied in a Mexican sample [32].

The translation into several languages can provide a precise way of measuring satisfaction/frustration of a person´s BPN, and it also receives key information about functioning. Thus, the translation and adaptation of tests is one of the concerns in psychometric research [33], since it can facilitate comparisons across cultures, and help understand variations in diverse cultures.

In another hand, eudaimonic well-being and SWB are closely related, and each one expresses different approaches to well-being; they both constitute genuinely distinct ways of understanding what "feeling well" implies [16]. Keyes [34] suggested that the SWB indicators should complement indicators of eudaimonic well-being (as psychological functioning) to identify if subjects are persons who function fully [1], as SWB only responds to the question of how the person feels, but not to the question of why the person feels that way [2]. Thus, is not only important to measure if people "feel well" (i.e., their subjective well-being); it is necessary to know why people feel well.

BPN satisfaction can be seen as the nucleus of eudaimonic functioning [2], considered as a key precedent of subjective well-being. In sum, Bradley and Crowyn [35] point out that life satisfaction reflects the way in which BPN are satisfied.

The research within the SDT framework has demonstrated that the needs of autonomy, competence and relatedness are related with several well-being indicators [36,37]. Hence, the people that find their BPN satisfied will display a more evident integration process and will tend to exhibit great well-being and life satisfaction [38], while subjective vitality is related with subjective well-being indicators, such as life satisfaction [13,39].

Additionally, there have been previous studies where significant differences are observed in the levels of both eudaimonic well-being as well as SWB by sex [40]. Thus, the factorial invariance analysis can provide an indication of whether the statistically significant differences of the scale scores can reflect real differences across the groups in the latent variables that are being measured.

Based on the aforementioned studies, Study 1 aims to prove the validity (from factorial structure, reliability and factorial invariance by sex) of a translation and adaptation of the BPNSFS into Spanish. The Study 2 has the purpose of proving a sequential theory model of the relation between satisfied and frustrated psychological needs (antecedents), subjective vitality (mediator) and life satisfaction (consequences) by sex, controlling for the age effect and analyzing subjective vitality mediator effect.

*2.1. Study 1*

2.1.1. Method

This is a study with an associative strategy with cross-sectional and observable variables, and comparative natural groups design [41].

Participants

A convenience sample was applied in the selection of participants, who were 830 University students of physical activity and sports from two Public State Universities in Mexico, from the northeast and northwest. The instrument was answered by students who attended the session, by registered students in the bachelor program (physical activity and sport) and by students who wanted to participate.

Age range was from 17 to 31 years (*M* = 20.70 years; ±2.96). Of the sample, 67.7% were men, the rest were women. To be able to reach the objectives, the sample was divided into two subsamples by a random selection of 50% of the cases, preserving the relative distribution of sex and age. This first subsample was made up of 415 students (280 men; 135 women).

Instruments

To measure the satisfaction and frustration of BPN, the Basic Psychological Need Satisfaction and Frustration Scale (BPNSFS) [24] was used. It is made up of 24 items grouped in six factors corresponding to the satisfaction of the needs of competence, autonomy and relatedness, and the frustration of the needs of competence, autonomy and relatedness. Each one of the factors is made up of four items which are answered with a Likert scale of 5 points that proceed from 1 (completely false) to 5 (completely true).

Procedure

The ethical approval to carry out this study was granted by the Ethics Committee of the University (UABC-1149) to the lead researcher. The established APA anonymity and confidentiality guidelines of the information were followed.

The English version of the BPNSFS was translated following the inverse translation procedure [42]. The items were translated into Spanish, and then translated again into English by a group of translators, observing similarities with the original version. The battery was evaluated by specialists in sport psychology who evaluated the relevance of its items regarding the construct measurement, as well as its correct wording. Next, it was applied to a small group of students to verify if the battery of questions was understood and to make the necessary corrections. After having the instrument´s final version in Spanish, and after receiving the permission of both University Faculties, a direct link to the online survey in Google Forms was provided. The students who decided to participate completed the multi-section test online. The data collection pilot procedure indicated that it took around 15 min to complete the survey. The order in which the measures were presented to the students was the same. The questionnaires were answered anonymously and voluntarily.

Data Analysis

Firstly, the data was examined with the SPSS 23 program to detect any missing values, normality and the presence of outliers. Next, to validate the factorial structure of the instrument, a confirmatory factor analysis (CFA) was conducted with the AMOS 26 program and maximum likelihood method.

For the CFA, the data were analyzed using a model with first-order factors which include six latent factors, corresponding to each one of the three BPN within the frustration and satisfaction components. Since SDT assumes that the three BPN coexist [6], a model with first-order factors was subsequently tested with constructs comprising need satisfaction and need frustration as two second-order factors.

Incremental, absolute and parsimonious adjustment indices were used to evaluate the model. Including RMSEA and its confidence interval at 90% (CI90), RSMR, TLI, CFI and PCFI. Values equal or lower than 0.08 for RMSEA indicate a good adjustment [43], with values equal or lower than 0.10 for the upper limit of the CI90 [44]. For the RMSR, values equal or lower than 0.08—and, for the CFI and TLI, values higher than 0.90—indicate an

acceptable adjustment [44]. As for the PCFI, the range goes from 0 to 1, where 1 reflects a perfect adjustment.

The equivalence of the instrument with a multigroup CFA was also tested to prove the factorial invariance by sex. Differences no greater than 0.01 for the CFI indicate irrelevant practical differences [45]; increments in RMSEA lower than 0.015 can support invariance [46].

Further, the composed reliability was analyzed with the coefficient McDonald´s omega, where values greater than 0.70 show a good reliability [47]. The average variance extracted (AVE) was tested, where values greater than 0.50 indicate a good adjustment [47].

### 2.1.2. Results

The preliminary analysis of the data suggested the absence of missing data and outliers. On the other hand, the first CFA confirmed the structure of six first-order factors of the instrument since the model adjustment was good (Table 1). All the items saturated significantly at the $p < 0.01$ level, with factorial weights greater than 0.50, except three items which saturated below the criterion (Table 2). The modification indices did not assume an improvement to the adjustment of the model by eliminating said items, while the phi matrix correlation displayed high correlations between the latent factors (Table 3)—mainly between competence satisfaction and autonomy satisfaction (phi = 0.87)—suggesting a lack discrimination among both factors, which is why a model with two second-order factors was tested; one of them grouped the satisfaction of the three BPN, named need satisfaction; the other factor grouped the frustration of the three BPN, named need frustration.

**Table 1.** Fit indices of tested models.

| Model | $\chi^2$ | *df* | RMSEA (IC 90) | TLI | CFI | RMSR | PCFI |
|---|---|---|---|---|---|---|---|
| Six factors model | 440.75 * | 237 | 0.046 (0.039–0.052) | 0.93 | 0.94 | 0.04 | 0.80 |
| Six factor model plus two second-order factors | 546.19 * | 245 | 0.055 (0.048–0.061) | 0.90 | 0.91 | 0.05 | 0.80 |

Note. * $p = 0.000$.

**Table 2.** Asymmetry, kurtosis and factorial weights of the items that make up the BPNSFS.

| Item | Asymmetry | Kurtosis | $\delta$ | $\lambda$ | $R^2$ |
|---|---|---|---|---|---|
| Autonomy satisfaction | | | | | |
| 1 I feel a sense of choice and freedom in the things I undertake [Tengo una sensación de decisión y libertad en las cosas que emprendo] | −1.04 | 1.21 | 0.29 | 0.53 | 0.28 |
| 7 I feel that my decisions reflect what I really want [Siento que mis decisiones reflejan lo que realmente quiero] | −0.70 | 0.08 | 0.31 | 0.57 | 0.32 |
| 13 I feel my choices express who I really am [Siento que mis elecciones expresan quien realmente soy] | −0.66 | 0.08 | 0.16 | 0.37 | 0.14 |
| 19 I feel I have been doing what really interests me [Siento que hago lo que en realidad me interesa] | −1.23 | 0.86 | 0.38 | 0.63 | 0.40 |
| Competence satisfaction | | | | | |
| 5 I feel confident that I can do things well [Me siento seguro de poder hacer las cosas bien] | −1.12 | 0.85 | 0.56 | 0.75 | 0.56 |
| 11 I feel capable at what I do [Me siento capaz en lo que hago] | −1.36 | 1.70 | 0.57 | 0.76 | 0.58 |
| 17 I feel competent to achieve my goals [Me siento competente para alcanzar mis objetivos] | −1.24 | 0.85 | 0.30 | 0.54 | 0.30 |
| 23 I feel I can successfully complete difficult tasks [Siento que puedo completar con éxito tareas difíciles] | −0.83 | 0.52 | 0.40 | 0.63 | 0.40 |
| Relatedness satisfaction | | | | | |

**Table 2.** *Cont.*

| Item | Asymmetry | Kurtosis | δ | λ | R² |
|---|---|---|---|---|---|
| 3 I feel that the people I care about also care about me [Siento que las personas que me importan también se preocupan por mí] | −1.14 | 0.62 | 0.39 | 0.62 | 0.38 |
| 9 I feel connected with people who care for me, and for whom I care [Siento conexión con las personas que se preocupan por mí y por las que yo me preocupo] | −1.16 | 0.87 | 0.49 | 0.70 | 0.50 |
| 15 I feel close and connected with other people who are important to me [Me siento cercano y unido a otras personas que son importantes para mí] | −0.96 | 0.21 | 0.56 | 0.75 | 0.56 |
| 21 I experience a warm feeling with the people I spend time with [Siento afecto hacia las personas con las que paso el tiempo] | −1.20 | 0.82 | 0.17 | 0.38 | 0.14 |
| Autonomy frustration | | | | | |
| 2 Most of the things I do feel like "I have to" [La mayoría de las cosas que hago se sienten como que "tengo que hacerlo"] | −0.37 | −0.43 | 0.15 | 0.36 | 0.13 |
| 8 I feel forced to do many things I wouldn't choose to do [Me siento obligado a hacer cosas que elegiría no hacer] | 0.68 | −0.44 | 0.50 | 0.71 | 0.50 |
| 14 I feel pressured to do too many things [Me siento presionado para hacer muchas cosas] | 0.17 | −1.07 | 0.50 | 0.71 | 0.51 |
| 20 My daily activities feel like a chain of obligations [Mis actividades diarias se sienten como una serie de obligaciones] | 0.27 | −0.78 | 0.47 | 0.69 | 0.48 |
| Competence frustration | | | | | |
| 6 I have serious doubts about whether I can do things well [Constantemente dudo de mi capacidad para hacer las cosas bien] | 0.13 | 1.12 | 0.37 | 0.60 | 0.36 |
| 12 I feel disappointed with many of my performance [Me siento decepcionado con mi desempeño en general] | 0.74 | −0.33 | 0.58 | 0.76 | 0.58 |
| 18 I feel insecure about my abilities [Me siento inseguro de mis habilidades] | 0.61 | −0.71 | 0.50 | 0.71 | 0.50 |
| 24 I feel like a failure because of the mistakes I make [Siento que soy un fracaso por los errores que cometo] | 1.02 | 0.13 | 0.60 | 0.77 | 0.60 |
| Relatedness frustration | | | | | |
| 4 I feel excluded from the group I want to belong to [Me siento excluido del grupo al que quiero pertenecer] | 1.04 | 0.03 | 0.88 | 0.59 | 0.38 |
| 10 I feel that people who are important to me are cold and distant towards me [Siento que las personas que son importantes para mí, son frías y distantes conmigo] | 0.95 | −0.05 | 0.72 | 0.71 | 0.50 |
| 16 I have the impression that people I spend time with dislike me [Tengo la impresión de que le desagrado a las personas con las que paso el tiempo] | 0.86 | −0.42 | 0.90 | 0.65 | 0.43 |
| 22 I feel the relationships I have are just superficial [Siento que las relaciones que tengo son sólo superficiales] | 0.68 | −0.37 | 0.72 | 0.67 | 0.44 |

Note. All factorial weights are significant at $p < 0.01$; δ = standard deviation; λ = factorial weights.

**Table 3.** Matrix of phi correlations between the first-order latent factors, AVE values and composed reliability of the factors of the BPNSFS.

| | AVE | Reliability | 1 | 2 | 3 | 4 | 5 | 6 |
|---|---|---|---|---|---|---|---|---|
| Autonomy satisfaction | 0.28 | 0.79 | | 0.87 | 0.72 | −0.51 | −0.63 | −0.38 |
| Competence satisfaction | 0.50 | 0.79 | | | 0.57 | −0.52 | −0.79 | −0.48 |
| Relatedness satisfaction | 0.39 | 0.78 | | | | −0.39 | −0.48 | −0.59 |
| Autonomy frustration | 0.40 | 0.79 | | | | | 0.77 | 0.70 |
| Competence frustration | 0.50 | 0.79 | | | | | | 0.80 |
| Relatedness frustration | 0.43 | 0.79 | | | | | | |
| Need satisfaction | 0.37 | 0.91 | | | | | | |
| Need frustration | 0.44 | 0.92 | | | | | | |

Note: all correlation coefficients are significant at $p < 0.05$.

The model with two second-order factors and six first-order factors, also showed that the measurement model adjusts well to the data (see Table 1). The factorial weights that represent the relations between latent variables showed positive and significant gamma coefficients of 0.93, 0.94 and 0.66 for the satisfaction of autonomy, competence, and relatedness, respectively, in their association with the latent variable of need satisfaction, while showing positive and significant gamma values of 0.78, 1 and 0.80 for the frustration of autonomy, competence and relatedness, respectively, in their association with the latent variable of need frustration.

The difference in RMSEA between both models suggested irrelevant practical differences ($\Delta$RMSEA = 0.009), while the difference in CFI suggested a better adjustment for the six-factor model ($\Delta$CFI = 0.03). The composed reliability coefficients of the scales were satisfactory (omega range of McDonald of 0.78 to 0.79). Nevertheless, the AVE values only supported the adjustment for the frustration and satisfaction factors of competence (Table 3).

To analyze the factorial invariance by sex, a series of multi-sample CFAs was carried out, which indicated that the first-order structure of the instrument is invariant between men and women; the comparisons of adjustment indices between models nested with restrictions confirmed the equivalences in the four models, supporting the strict invariance. The same thing was observed for the model of two second-order factors (Table 4).

**Table 4.** Fit indices of each of the models tested in the factorial invariance of the BPNSFS between men and women.

| Models | | $\chi^2$ | *df* | RMSEA | CFI | $\Delta$RMSEA | $\Delta$CFI |
|---|---|---|---|---|---|---|---|
| First-order six-factor model | | | | | | | |
| M1 | Unconstrained model | 784.46 | 474 | 0.040 | 0.909 | | |
| M2 | Factorial weights constrained | 797.5 | 492 | 0.039 | 0.911 | 0.001 | 0.002 |
| M3 | Factorial weights and intercepts constrained | 837.76 | 513 | 0.039 | 0.905 | 0.001 | 0.004 |
| M4 | Factorial weights, intercepts and error variances constrained | 874.68 | 537 | 0.039 | 0.901 | 0.001 | 0.008 |
| Model of six first-order factors plus two second-order factors | | | | | | | |
| M1 | Unconstrained model | 784.46 | 474 | 0.040 | 0.909 | | |
| M2 | Factorial weights constrained | 797.05 | 492 | 0.039 | 0.911 | 0.001 | 0.002 |
| M3 | Factorial weights and intercepts constrained | 837.76 | 513 | 0.039 | 0.905 | 0.001 | 0.004 |
| M4 | Factorial weights, intercepts and error variances constrained | 874.68 | 537 | 0.039 | 0.901 | 0.001 | 0.008 |

### 2.1.3. Brief Discussion

This first study was conducted to prove the validity (from factorial structure, reliability and factorial invariance by sex) of a translation and adaptation of the BPNSFS into Spanish. The factorial structure of six first-order factors (i.e., satisfaction of autonomy, satisfaction of competence, satisfaction of relatedness, frustration of autonomy, frustration of competence and frustration of relatedness) is confirmed. As well as the six first-order factors, there are, in addition, two second-order factors (i.e., need satisfaction, need frustration). This results match with other studies (e.g., [26–28,32]).

Although three items had a factorial weight under the criterion, their factorial weights are significant and their elimination does not provide an improvement to the model. However, removing them does result in an improvement to the validity, since the AVE displays an acceptable level of convergent validity and compound reliability when all the standardized regression coefficients of a latent variable are significant and higher than 0.50, even if its AVE is under 0.50 [43]. These three items also displayed low factorial weights in other linguistic adaptations (e.g., [26–28]).

Regarding the discriminant validity of the first-order factors, satisfaction of competence and satisfaction of autonomy correlated very highly, which is oppositional to said

validity; nevertheless, the same can be seen in other adaptations of the instrument, as in Kuzma et al. [48] and Zamarripa et al. [32]. Likewise, the strict invariance of the scale by sex is confirmed. These psychometric properties are added to the studies of Cardella et al. [31] and Zamarripa et al. [32].

This first study validates an instrument of measurement of satisfaction and frustration of BPN. Furthermore, it guarantees that the items capture the exact meaning of the three BPN based on the SDT, where satisfaction of competence captures the degree in which people feel efficient in their interactions with the environment and experiment with opportunities to demonstrate their capabilities [49]; the satisfaction of autonomy captures the sensation of freedom to be oneself and take one´s own decisions [9]; the satisfaction of relation captures the feeling of closeness and relation with others [50]. The frustration of relation encloses the experience of exclusion and isolation; the frustration of competence encloses feelings of failure and preoccupation about one´s efficiency; and the frustration of autonomy encloses feelings of control through external forces or self-imposed pressures [6].

Since the BPN category provides a parsimonious set of elements and the center of the well-being construct [2], this BPNSFS adaptation can be used in attempts to measure functioning eudaimonic aspects in a more comprehensible manner.

### 2.2. *Study 2*

This study is an extension from Study 1. Here, interest in the BNP frustration and satisfaction consequences is for pragmatic and conceptual reasons.

We study the hypothesis that when people have high autonomous tendencies and fewer internal conflicts, and they trust their abilities to carry out the suggested tasks, they experiment more vitality and this leads to more life satisfaction. The opposite happens if the BPN are frustrated (see Figure 1).

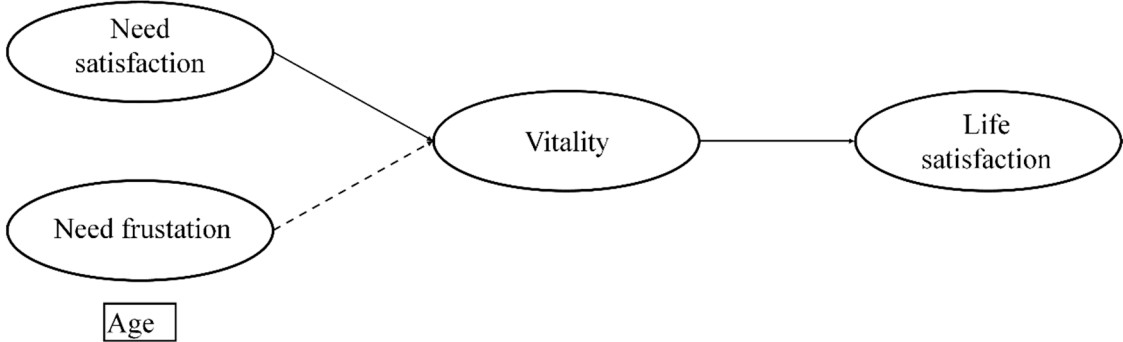

**Figure 1.** Hypothesized model with expected relationships. The dashed line represents negative association.

### 2.2.1. Method

Participants

The second subsample consisted of 415 university students (281 men; 134 women) ranging in ages from 17 to 31 years old (M = 20.70 years; SD = 2.96).

Instruments

Needs satisfaction and frustration were measured with the Spanish version of the Basic Psychological Need Satisfaction and Frustration Scale described in Study 1.

Levels of subjective vitality were measured with the Spanish version of the Subjective Vitality Scale (SVS) [51]. It consists of six items (e.g., "I have energy and mood") with a 7-point Likert-type response format ranging from 1 (not true at all) to 7 (very true).

The Spanish version of the Satisfaction With Life Scale (SWLS) [52] was used to measure the degree of satisfaction with one's own life. It consists of five items (e.g., "The

circumstances of my life are good") that are answered with a 5-point Likert scale ranging from 1 (strongly disagree) to 5 (strongly agree).

Data Analysis

First, descriptive and bivariate correlation statistics were calculated for the study variables using the SPSS 23 software. After that, the measure models were tested by CFA. Given the evidence of scalar factorial invariance between the sex of the measurement instruments used, this allows comparison of the relationships between the latent variables between the two groups. For this reason, a multi-group structural equation model (MSEM) was developed to examine associations between NPB satisfaction and frustration, with vitality and life satisfaction outcomes across sex, controlling for the effect of age. The two analysis steps proposed by Wang et al. [53] were followed; they comprise the testing of a model correlating the latent variables, and then a structural equation model (MSEM) with the need frustration composite variable and the need satisfaction composite variable as antecedents of subjective vitality, and life satisfaction as a consequence.

The MSEM was supplemented by bootstrapping analysis (5000 samples), as recommended by Preacher and Hayes [54], and percentile confidence intervals al 95% to determine unstandardized values and significance levels for indirect effects in the hypothesized model. Indirect effects were considered statistically significant if the 95% confidence interval did not include zero [55].

Both the CFAs and MSEM were analyzed with the AMOS 26 software, using the maximum likelihood estimation method. Dince the multivariate Mardia's coefficient result was 61.40, this indicates a multivariate non-normality of the data. Nevertheless, the univariate normality analysis showed that data were closed to normality (asymmetry and kurtosis between −2 and 2).

### 2.2.2. Results

As for the preliminary analyses, Table 5 shows the means, standard deviation and bivariate correlations among the variables. The satisfactions of each of the three needs were positively related to each other, and each of these three was related to subjective vitality and life satisfaction. While frustrations of each of the three needs were positively related to each other, and negatively related to subjective vitality and life satisfaction.

**Table 5.** Descriptive statistics and Pearson's correlation matrix between the study variables.

| | **Satisfaction** | | | **Frustration** | | | **Vitality** | **Life Satisfaction** |
|---|---|---|---|---|---|---|---|---|
| | **Autonomy** | **Competence** | **Relatedness** | **Autonomy** | **Competence** | **Relatedness** | | |
| 1 | | | | | | | | |
| 2 | 0.60 ** | | | | | | | |
| 3 | 0.46 ** | 0.43 ** | | | | | | |
| 4 | −0.29 ** | −0.39 ** | −0.27 ** | | | | | |
| 5 | −0.41 ** | −0.62 ** | −0.27 ** | 0.60 ** | | | | |
| 6 | −0.25 ** | −0.37 ** | −0.44 ** | 0.52 ** | 0.62 ** | | | |
| 7 | 0.53 ** | 0.64 ** | 0.37 ** | −0.39 ** | −0.50 ** | −0.33 ** | | |
| 8 | 0.53 ** | 0.59 ** | 0.39 ** | −0.38 ** | −0.47 ** | −0.34 ** | 0.69 ** | |
| M | 4.08 | 4.20 | 4.23 | 2.76 | 2.28 | 2.06 | 5.46 | 3.84 |
| DT | 0.63 | 0.70 | 0.68 | 0.88 | 0.96 | 0.90 | 1.22 | 0.81 |

Note. ** $p < 0.001$; M = mean; SD = standard deviation.

The AFCs of the measurement models provided good fit for each instrument. For the instrument measuring Subjective Vitality, the unifactorial structure was confirmed with acceptable fit indices: $\chi^2 = 25.17(8)$, $p = 0.001$; RMSEA = 0.07 (CI90 0.04–0.10); CFI = 0.98; TLI = 0.97. For the instrument measuring life satisfaction, the unifactorial structure was confirmed with acceptable fit indices: $\chi^2 = 5.92(4)$, $p = 0.205$; RMSEA = 0.03 (CI90 0.00–0.08); CFI = 0.99; TLI = 0.99.

Structural Equation Models with Multiple Group Analysis (SEM Multi-Group)

As for the preliminary model correlating the latent variables, it showed good fit: $\chi^2$ = 386.287(113), $p$ = 0.000; RMSEA = 0.07 (CI90% 0.06–0.08); CFI = 0.93; TLI = 0.91. This allowed the testing of the hypothesized model, which exhibited adequate fit to the data: $\chi^2$ = 607.96(256), $p$ = 0.000; RMSEA = 0.05 (CI90% 0.05–0.06); CFI = 0.92; TLI = 0.90. The standardized parameter estimates of the effects are presented in Figure 2. The results indicated that, for both women and men, need satisfaction has a positive effect on subjective vitality, and subjective vitality is positively associated to life satisfaction. Need frustration had no significant effect on subjective vitality. Subjective vitality explained 55% and 80% of the variance in men and women, respectively, while life satisfaction explained 66% and 75% of the variance.

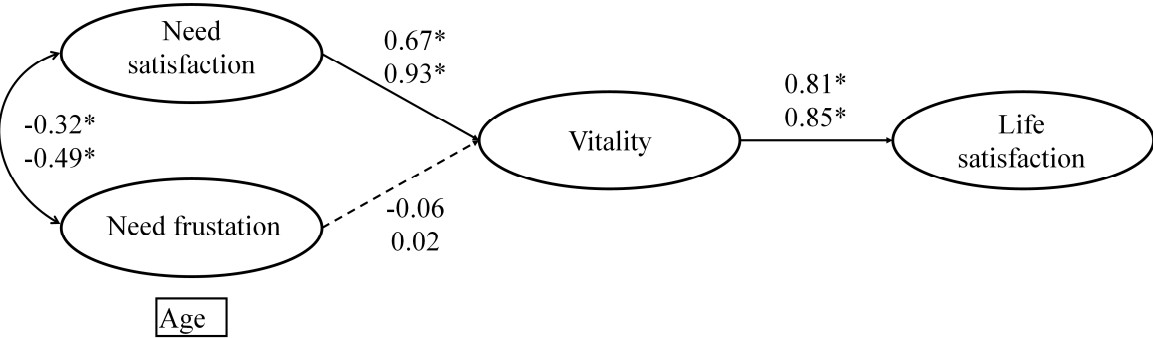

**Figure 2.** Structural relationships between NPB satisfaction and frustration, subjective vitality and life satisfaction. Upper values—men; down values—women. * $p$ = 0.000.

To explore whether this MSEM model of total mediation is equally applicable in both men and women, a multi-group model was estimated according to sex. The results of the multi-group SEM suggested that the structure of the proposed model is equivalent for men and women, so that the correlates of psychological needs and well-being were statistically equivalent between men and women, as the unconstrained model fitted satisfactorily and the difference in the goodness-of-fit indices between the constrained models nested in the unconstrained model were trivial (Table 6).

**Table 6.** Goodness-of-fit indices for each of the models tested on the structural relationships between men and women.

| Nested Models | $\chi^2$ | $gl$ | RMSEA | CFI | ΔRMSEA | ΔCFI |
|---|---|---|---|---|---|---|
| Unrestricted model | 607.96 | 256 | 0.058 | 0.911 | | |
| Measurement weights | 630.62 | 269 | 0.057 | 0.909 | 0.001 | 0.002 |
| Measurement intercepts | 686.84 | 286 | 0.058 | 0.900 | 0.000 | 0.011 |
| Structural weights | 694.47 | 291 | 0.058 | 0.900 | 0.000 | 0.011 |
| Structural means | 694.47 | 292 | 0.058 | 0.900 | 0.000 | 0.011 |
| Structural covariances | 714.19 | 298 | 0.058 | 0.899 | 0.000 | 0.012 |
| Structural residuals | 714.96 | 300 | 0.058 | 0.899 | 0.000 | 0.012 |
| Measurement residuals | 746.21 | 317 | 0.057 | 0.895 | 0.001 | 0.016 |

The calculation of the mediation effects of subjective vitality indicated that said variable mediates the relationship between psychological need satisfaction and life satisfaction, since the 95% confidence interval of the indirect effect did not include zero (Table 7). Contrary to this, the relationship between need frustration and life satisfaction was not mediated because the confidence interval crossed to zero.

**Table 7.** Indirect effects of the mediation model of the effects of NPB satisfaction and frustration on life satisfaction, with subjective vitality as mediator.

| Independent Variable | ab Coefficients Product | CI 95% |
|:---:|:---:|:---:|
| Men | | |
| Need satisfaction | 0.53 * | 0.35 to 0.74 |
| Need frustration | −0.04 * | −0.13 to 0.03 |
| Women | | |
| Need satisfaction | 0.81 * | 0.44 to 1.71 |
| Need frustration | 0.01 * | −0.26 to 0.63 |

Note. * $p < 0.05$.

### 2.2.3. Brief Discussion

Study 2 had the purpose of proving a sequential theory model of the relation between satisfied and frustrated psychological needs (antecedents), subjective vitality (mediator) and life satisfaction (consequences) by sex, controlling for the age and subjective vitality mediator effects. In this sample of university students of physical activity and sports, the first part of the MSEM confirms that the experience of personal initiative and a greater sense of effectiveness and understanding explain the conscious experience of feeling good, i.e., a general energy for life. This is consistent with deCharms' [56] proposal: as long as one is free of conflicts and external controls and feels able to act, then one should report a greater experience of oneself as a potential "source." This also agrees with the study of Chen et al. [24].

On the other hand, it is not confirmed that the experience of incompetence, being rejected by others and controlled by external forces decreases subjective vitality. This suggests that need frustration may be a better predictor of ill-being [12], as lack and need frustration should be consistently and directly associated with indicators of ill-being [57]. Such results are consistent with the study by Chen et al. [24]. It is supported that people differ in their subjective vitality as a function of NPB satisfaction or frustration [13].

On the other hand, the second part of the model confirms that a positive and accessible state of acquiring energy for oneself [13] is positively related to individuals' overall judgment of how satisfied they are with their life [58], as suggested by Ryan and Frederick [13] and agreeing with other studies (e.g., [16,39]).

Mediation analysis supports that those who report high NPB satisfaction feel better about their own lives in general (life satisfaction [59], as measure through subjective vitality. So, life satisfaction reflects how NPBs are satisfied [35] as long as it is aligned with self-values and needs. Therefore, the reason why people feel good is because of the antecedent that they function with feelings of support, efficacy and relatedness, thus providing information about their optimal function, which is invariant between men and women.

### 3. General Discussion

SWB plays a key role in the subsequent adaptive functioning of individuals, being an indicator of societal health and a predictor of future functioning. Beutell [60] believes that life satisfaction is related to better physical and mental health, longevity and other outcomes that are considered to be positive in nature. Therefore, the importance of knowing how satisfied with life a person is lies in the fact that those who experience high life satisfaction have energy, enthusiasm and liveliness [39]; furthermore, high life satisfaction suggests that one's quality of life is good. Neal et al. [61] considers that life satisfaction is functional and related to satisfaction in all domains and subdomains of life, suggesting that satisfaction in one domain of an individual's life extends to other areas.

Based on Keyes [5], we can say that people who score high on eudaimonic indicators (BPN and subjective vitality) and SWB (satisfaction with life) have better mental health than other individuals. However, it is important to consider that hedonia (with indicators such as life satisfaction) can lead to addiction, chronic escapism, dangerous impulsive behavior,

selfishness, antisocial behavior, greed, excessive consumerism, etc. Eudaimonia (with indicators such as BPN and subjective vitality) can result in a workaholic lifestyle, burnout, excessive self-sacrifice, overthinking things, over-theorizing and losing practicality [62].

It is usually supported that BPNs pertain to eudaimonic well-being because they contribute to SWB [2]; thus, these may provide the common core for eudaimonic indicators of well-being [2]. The distinction between BPN frustration and satisfaction seems useful, as both constructs have unique associations with subjective vitality and life satisfaction.

Among the results, such as those of Bartholomew et al. [12], Chen et al. [24] and Zamarripa et al. [32], a negative relationship between need satisfaction and need frustration is observed.

This study has theoretical and practical contributions. From a theoretical point of view, it contributes to the construct validation of a scale that measures satisfaction and frustration of the three BPNs in another country. In addition, we provide evidence for the scale measurement equivalence, suggesting that the items included are understood similarly for both men and women. Thus, we provide support for the use of the instrument to measure differences in satisfaction and frustration between men and women, and for comparison of the observed scores.

Overall, measuring BPN gives us a broader view of people's psychological functioning and what makes them feel good [22], helping to identify key ways to improve well-being.

Within the practical implications, the Spanish version of the BPNSFS will provide the Mexican scientific community with a valid and reliable instrument with which to measure the level to which BPNs are satisfied or frustrated in the general context, alongside the adaptation to physical education (i.e., [32]). This can support college students' health; they represent national assets and a future investment for society [63–65]. It is therefore crucial to understand college students' life satisfaction, discover how to promote it and prevent psychopathologies.

Although, this instrument allows comparison across cultures, it should be noted that the validation of a scale must have continuity over time using different samples and more studies to verify the validity and reliability of these results; it is therefore suggested that future studies should be carried out in different sociodemographic contexts in order to determine the real usefulness of this scale.

The differences in how people perceive frustration or satisfaction of needs in a particular context is important, so we suggest the application of the instrument in other contexts and other Spanish-speaking countries.

The convenience sample limits the study's representativeness of the population. Therefore, we should be cautious in generalizing the present results. Another important limitation is the possible existence of common-method variance by self-report data collection. However, both the size of the two study subsamples, and the reliability contrasted in these, consistently demonstrate the remarkable psychometric characteristics of the scale.

Another limitation was the cross-sectional design, which does not allow causal conclusions to be drawn. Overall, in future research, we suggest the inclusion of consequences of ill-being, and the separate examination of how each need contributes to well-being.

## 4. Conclusions

BPN satisfaction is an essential aspect to address since it functions as a fundamental nutrient that contributes to the perception of well-being; in this sample, the more satisfied are university physical activity and sports students' needs, the more vital they look and, consequently, the more satisfied they are with their lives, which is invariant between men and women. Lastly, it is supported that a state of life satisfaction is impossible without BPN satisfaction [22].

**Author Contributions:** H.A.P.-E., R.M.-S., L.T., I.L.-G., A.H.-M., V.M.-S., E.C.-M. and A.G.-G. participated in the study design and data collection, performed statistical analyses and contributed to the interpretation of the results, wrote the manuscript and approved the final manuscript as presented. All authors made substantial contributions to the final manuscript. All authors have read and agreed to the published version of the manuscript.

**Funding:** This research was funded by Universidad Autónoma de Baja California grant number [2022].

**Institutional Review Board Statement:** The study was conducted in accordance with the Declaration of Helsinki, and approved by the Ethics Committee of Universidad Autónoma de Baja California (protocol code 1149).

**Informed Consent Statement:** Informed consent was obtained from all subjects involved in the study.

**Data Availability Statement:** Data supporting reported results can be requested from the main author.

**Conflicts of Interest:** The authors declare no conflict of interest.

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
