# Peer review of "Satisfied and Frustrated Needs, Subjective Vitality and University Students’ Life Satisfaction of Physical Activity and Sports"

_sustainability, doi:10.3390/su15043053_

Round 1

Reviewer 1 Report

Dear Editor, Thank you for selecting me to review the manuscript entitled "Indicators of eudaimonic and subjective well-being in physical activity Mexican students". The article shows the results of two studies in university students of physical activity and sport in Mexico. The article is well written and the methodology and results are of general interest. In the first study the adaptation of the Basic Psychological Need Satisfaction and Frustration Scale is translated and validated in Spanish. The results of the CFA show adequate fit indices. In the second study, through a structural equation model, the mediational relationship between psychological need satisfaction and frustration on eudaimonic well-being and subjective well-being is analyzed. However, several concerns arise in this case: Q1. Why was the age of the participants not entered in the models as a variable? Are the results expected to be similar across the age range of the sample? Q2. Further explanation is needed, with previous studies, of the differences between the concepts of subjective vitality and life satisfaction, as well as why subjective vitality and life satisfaction are measured as proxies for eudaimonic well-being and SWB.

Author Response

We appreciate the opportunity to review our manuscript in detail. We greatly appreciate your insightful comments, and we are convinced that the work has greatly benefited from your review.
In accordance with the suggested recommendations, we have made substantial changes to the document, considering each and every one of the suggestions indicated.

We sincerely appreciate your comment.

The age of the participants was entered as a control variable within the MSEM , as can be seen in Figure 2, and in the data analysis section. Now, this has also been mentioned in the objective, and in figure 1 to make it clearer.

In general therms, in the introduction it is mentioned that we rely on the eudaimonic activity model, specifically on the component of feel good. Here are aspects of eudaimonic well-being (includes indicators such as BPN), and subjective well-being (includes indicators such as satisfaction with life). We mention that Rodríguez-Carvajal et al. (2010) also support that life satisfaction is an experience of subjective well-being.
Moreaover, it is mentioned that Ryan and Deci (2001) consider subjective vitality as an indicator of eudaimonic well-being.
About explanations with previous studies, we mention studies like the one by Uysal et al. (2014), and the SEM is supported by other studies such as Çelik (2017) and Jackson and DiPlacido (2020).

Once again, we appreciate your valuable comments and we look forward to your review.

Reviewer 2 Report

My recommendations are the following:

Abstract- mention that the sample was divided into two parts, I recommend replacing the word parts into sub-samples. Also used as they have been - used in different approaches, I recommend replacing the word used with applied, experienced, etc.

The conclusion is much too generated, use the phrase - he optimal functioning of people, while the study has a limited sample in terms of age. I recommend rewriting the conclusion. I recommend a review of the abstract and attention to expression.

Keywords: I recommend a more concrete and study-focused choice of keywords, eg: multigroup and validity, they are not representative.

I recommend that bibliographic references be mentioned throughout the article in accordance with the editing rules.

I recommend mentioning the purpose of the study in the introduction.

Study 1, I recommend reformulating the title of this subsection. Also the union of Introductions and Study 1, in a single section and its revision.

In the introduction, do not mention anything about study 2.

Searching through the article I noticed that you mention them as two distinct parts, which makes it difficult to understand, I recommend that you restructure and organize the article.

SD= 2.96 is written using ±, I recommend correction.

I recommend rewriting - A convenience sample was used in the selection of participants, it is not clear what the inclusion and exclusion criteria were in the group, convenience does not specify anything.

In the Tools section, I recommend mentioning the value of Chronbach's α for the entire questionnaire and for the factors.

Procedure - mention the number of the approval decision of the ethics committee.

I recommend that the Discussion section of study 1 be expanded with new correlations of the results obtained with results from previous studies.

I recommend that the Data analysis sections be more concise.

Table 5 what does DT represent?

The Conclusions section I recommend to be completely rewritten, it is general and unfocused.

The idea of this article does not have a significant contribution to academic knowledge, it is also quite difficult to follow. I recommend reorganizing and then resubmitting to another type of journal.

Author Response

We appreciate the opportunity to review our manuscript in detail. We greatly appreciate your insightful comments, and we are convinced that the work has greatly benefited from your review.
In accordance with the suggested recommendations, we have made substantial changes to the document, considering each and every one of the suggestions indicated.

The word used has been replaced by other synonyms.
The word parts was replaced by subsample

We greatly appreciate your recommendation. In this regard, we have tried to improve the conclusion as follows, hoping to address the observation:
“…in this sample, the most satisfied university physical activity and sports students’ needs, they look more vital and, consequently, the most satisfied they are with their lives, which is invariant between men and women.…”
In the abstract it now says “In conclusion, the competence, relationship and autonomy satisfaction are essential nutrients for a positive performance in this sample.”

The keywords are now basic psychological needs, psychometric properties, psychological well-being; sport.

Added a subtitle of Current study where the two introductions were joined in this section

Given the adjustments to the structure of the manuscript, the background and theoretical framework of study 2 are now mentioned in the subtitle current study, hoping with this to help the reader understand the text.
The article has been reorganized, so that there is an introduction, then current study (which includes background and theoretical framework of studies 1 and 2), and continues with the methodological section of study 1.

we correct SD

Within Participants, the following text was added: “The instrument was answered by students who attended the session, by registered students in the bachelor program (physical activity and sport), and by students who wanted to participate.”

Regarding reliability, we opted for the McDonald's omega value because of the confirmatory nature of the study. These values are presented in Table 3.

Now we mention the number of the approval decision of the ethics committee.

On page 8 you can read the suggested changes. For example, it has been added that the factorial structure “agrees with other studies (e.g. Abidin et al., 2021; Phuoc, 2020; Zamarripa et al., 2020; Zayed et al., 2021).”

Data analysis sections was revised to try be more concise.

DT replaced by SD

We have tried to improve the conclusion, hoping to address the observation.
In general, the structure of the article has been reorganized.

Once again, we appreciate your valuable comments and we look forward to your review.

Reviewer 3 Report

Abstract:

It is not customary to include references in the abstract. I suggest authors delete (BPNT; Deci & Ryan, 2000).

Keywords: I suggest that they write up to 56 keywords. They do not use words already in the title: Students; well-being.

Theoretical framework.

I congratulate the authors on the quality of their work. The manuscript incorporates two studies that complement each other and provide a global view of the phenomenon studied.

I want to suggest that the authors try to develop some of the following aspects in their theoretical framework:

The Mexican context. I think it would be exciting to know if any works are linked to the object of study in the context of Mexico. This aspect could reinforce the strength of this article.

Gender variable. Gender is one of the variables analysed in the different statistical tests. However, in the theoretical framework and also in the discussion, the authors do not make any consideration of it. I suggest that you consider developing this variable in both sections.

Method.

I propose to the authors that the method section should begin by identifying which type of study corresponds to this research in the different studies. (cf. Ato, Manuel; López, Juan J.; Benavente, Ana Un sistema de clasificación de los diseños de investigación en psicología. Anales de Psicología, vol. 29, núm. 3, octubre, 2013, pp. 1038-1059.

Discussion

I suggest to the authors that the discussion section in the different studies should begin by recalling the objectives that were intended to be studied.

Limitations.

I would like to know whether the authors think that the sporting history of the students of physical activity and sports should have been considered. The results could be different depending on the type of physical activity performed and the level of competition.

I congratulate the authors again for this work and thank them for the effort to incorporate the suggestions made.

Congratulations again to the authors for this research.

Author Response

We appreciate the opportunity to review our manuscript in detail. We greatly appreciate your insightful comments, and we are convinced that the work has greatly benefited from your review.
In accordance with the suggested recommendations, we have made substantial changes to the document, considering each and every one of the suggestions indicated.

The citation that appeared in the abstract was removed.

The new keywords are basic psychological needs, psychometric properties, psychological well-being; sport.

Regarding the Mexican context, now it is mentioned that “There is also a Spanish version adapted in physical education and applied in a Mexican sample (Zamarripa et al., 2020).

Regarding the observation of the gender variable, this is mentioned both in the introduction and in the discussion on pages 3, 11 and 13.

Added study type, now says:
This is a study with an associative strategy and cross-sectional with observable variables, as well as a comparative with natural groups design (Ato et al., 2013).

Objectives were added to each brief discussion.

Regarding the type of physical activity performed and the level of competition, in this case it was a translation into Spanish and adaptation to the Mexican context of the BPNSFS, which measures needs in the general context, and not in the specific context of the sport. We carried out the validation process using a sample of university students in physical activity and sports. In this sense, it was not of interest to consider the sport practiced or the level of competition in the analysis.

Once again we sincerely appreciate all your comments, and we look forward to your review

Reviewer 4 Report

The main purpose of the work aimed to analyzed how the influences of satisfaction and frustration of basic psychological in the well-being of university physical activity and sport students, through sex.

The study is well designed. However I have some comments I’d like to express.

1. I suggest a title change to be more specific to the study

2. The abstract must be improved so that a good understanding of the readers.

3. Please clearly mention the results (statistical values) in the abstract.

4. Check key words based on Mesh standard.

5. References in the text and in the reference part are not according to Sustainability journal.

6. You have used old references (Veenhoven et al. 1993; Compton, Smith et al., 1996; Diener, 1984, 1999; Diener et al., 1985; Rodríguez-Carvajal et al., 2010; Ryff, 1989). Please change the references.

7. Define well-being and indicate its importance in sport (in the introduction)

8. More detail on the participants: How many years the subject's experience in physical activity?, number of training session/week?

9. In order to provide readers with a clearer understanding of the research, it is suggested that the author(s) may add a research structure diagram

10. More details for the conduct of the questionnaires

11. Please describe the abbreviations used in the tables

12. Conclusion: It is suggested that the author(s) can add future research directions or further research and verification issues.

Finally, please kind provided point-by-point responses highlighted in the text

Author Response

We appreciate the opportunity to review our manuscript in detail. We greatly appreciate your insightful comments, and we are convinced that the work has greatly benefited from your review.
In accordance with the suggested recommendations, we have made substantial changes to the document, considering each and every one of the suggestions indicated.

1. The title has been changed
Satisfaction and frustration need, subjective vitality and university students’ life satisfaction of physical activity and sports

2. We have tried to improve the abstract.

3. statistical values were added in the abstract.

4. Keywords have been updated
basic psychological needs, psychometric properties, psychological well-being; sport.

5. In our past experiences with Sustainability journal, we have been fortunate that, if the article is accepted for publication, the editorial team makes adjustments in citations and references according to the style of the journal itself. For this reason, we dared to send it in APA format, hoping that once again we will be lucky with the editorial process.

6. We appreciate the comment, and we have abandoned old references, staying with Veenhoven et al. 1993, given that it can be considered a classic, and with Rodríguez-Carvajal et al., 2010, which we believe is not such an old reference.

7. The following has been added:
Ryan and Deci (2001) establish to organize the studies well-being studies in two traditions. One is basically associated with happiness (hedonic will-being), and another related with the human potential development (eudaimonic will-being). Both are relevant in sports and physical activity, because they are required to plane objectives, set goals, to adapt to changes as well as, a high volition.

8. We have qualified that the sample was made up of university students in physical activity and sports, but not actually practicing sports, therefore this information was not considered in the data collection. We know that 23% of the sample did not practice sports.

9. In response to this suggestion and that of the other reviewers, the structure of the manuscript was modified, hoping to provide more clarity to the reader. There is an introduction, a subtitle of current study (here the antecedents of studies 1 and 2 appear).

11. All abbreviations in the tables have been described.

12. Suggested future research directions are reported on pages 13 and 14, just before the conclusion.

Once again we sincerely appreciate all your comments, and we look forward to your review

Round 2

Reviewer 2 Report

No comments

Reviewer 4 Report

Thank you for your effort

Authors responded well to comments